# Youtube^TM^ Content Analysis as a Means of Information in Oral Medicine: A Systematic Review of the Literature

**DOI:** 10.3390/ijerph19095451

**Published:** 2022-04-29

**Authors:** Antonio Romano, Fausto Fiori, Massimo Petruzzi, Fedora Della Vella, Rosario Serpico

**Affiliations:** 1Multidisciplinary Department of Medical-Surgical and Dental Specialties, University of Campania “Luigi Vanvitelli”, Via L. de Crecchio 6, 80138 Naples, Italy; fausto.fiori@outlook.com (F.F.); rosario.serpico@unicampania.it (R.S.); 2Interdisciplinary Department of Medicine, University of Bari “A. Moro”, 70124 Bari, Italy; massimo.petruzzi@uniba.it (M.P.); dellavellaf@gmail.com (F.D.V.)

**Keywords:** Youtube^TM^, social media, oral medicine, web medicine

## Abstract

Background: Oral medicine represents a complex branch of dentistry, involved in diagnosing and managing a wide range of disorders. Youtube^TM^ offers a huge source of information for users and patients affected by oral diseases. This systematic review aims to evaluate the reliability of Youtube^TM^ oral medicine-related content as a valid dissemination aid. Methods: The MeSH terms “Youtube^TM^” and “oral” have been searched by three search engines (PubMed, ISI Web of Science, and the Cochrane Library), and a systematic review has been performed; the PRISMA checklist has been followed in the search operations. Results: Initial results were 210. Ten studies definitely met our selection criteria. Conclusions: Youtube^TM^ represents a dynamic device capable of easy and rapid dissemination of medical-scientific content. Nevertheless, the most of information collected in the literature shows a lack of adequate knowledge and the need to utilize a peer-reviewing tool in order to avoid the spreading of misleading and dangerous content.

## 1. Introduction

Oral medicine is the specialty of medicine involved in the diagnosis and treatment of diseases, disorders, and conditions affecting the oral and maxillofacial region and in the oral health care of medically complex patients [1]. Oral medicine is interested in different diseases ranging from neurological affection [2], infectious diseases [3,4], chronic inflammatory disease [5], autoimmune diseases [6,7,8,9] and systemic diseases with oral manifestations [10].

Search engines and social media have revolutionized the relationship between the physician and the patients, with the patients more involved, due to the easier way to access health information, in the medical decisions concerning all of the diagnosis and treatment workflow [11]. The data on patients looking for health information on the internet in the period 2008–2010 ranged from 60% to 80%, showing an increase from 25% in 2000, with the main issue remaining the reliability of the information [12]. Google is one of the most used tools for medical information, and the use of Google Trends has already been proved as a useful mean to solve epidemiological points about disease tracking [13]. Google is a potentially easily accessible means of accessing scientific data, but the customization of search results basing on geographical aspects and previous search data, such as interests and interactions, makes the website an untrustworthy tool for medical information [14]. Among all the social media, Youtube^TM^ seems to be one of the most used platforms by patients for searching health-related information, with at least 2 billion views per day and a new video uploaded, on average, every minute [15]. Several authors assessed the quality of information spread via Youtube^TM^ for different medical branches highlighting that at least about 30% of the information was deemed as non-reliable, misleading, or dangerous due to the consumer-generated information and to the lack of a certified review for scientific content [16,17]. In particular, different studies have analyzed the content quality for oral healthcare information showing a darker picture on the reliability of information about dental medicine with a low-reliability group ranging from 60% to nearly 90% [18,19]. Nowadays, a gap seems to exist in knowledge regarding the reliability of health information video content, specifically in oral medicine and pathology. The purpose of this study was to systematically analyze all the papers about Youtube^TM^’s oral pathology items to underline all the pros and cons concerning this means of scientific divulgation.

## 2. Materials and Methods

A systematic literature search was performed using different database: PubMed, the Cochrane Library, and ISI Web of Science. Search strategy followed the PRISMA checklist, querying databases for “Oral” and “Youtube” as MeSH terms. The search operations on the research engine ended on 31 October 2021.

The selection criteria used to include the studies were: studies on Youtube^TM^ videos concerning oral pathologies, available full text, and published in English since 1990.

The selection was made in steps and by different operators: initially, all the articles were collected, then two reviewers (FF and AR) excluded duplication and eliminated articles that did not meet the inclusion criteria by reading the titles and abstracts of the studies. We decided to include all the abstracts strictly related to oral pathology and medicine. The full texts of the remaining works were evaluated further by two reviewers (AR and RS). Data extraction of the included papers has been synthetized in Table 1, in order to provide usable descriptive and technical information.

## 3. Results

The initial research yielded 210 studies; of these, 53 were duplicates. Of the 157 studies remaining, 133 were excluded by reading titles and abstracts as they did not meet the inclusion criteria. Most of the excluded abstracts was not strictly adherent to oral pathology and medicine issues since they were focused on other stomatological fields, such as endodontics, prosthodontics and implantology [20,21,22]. Reading the full texts, 14 other articles were excluded. In the end, 10 studies were identified (Figure 1). Information on the 10 selected articles and main video details such as oral pathology subject, search strategy, number of examined videos, video length and qualitative analysis tools are summarized in Table 1. A total number of 672 retrieved videos are included in the analysis of all the selected articles; mean video length ranges from 03:09 ± 01:48 to 09:03 min (a single paper [23] does not mention the collection of data relating to the length of the videos).

The main aim of this research focuses on oral medicine Youtube content, so that the workflow led us to exclude some papers concerning implant dentistry, which seems a hot topic for web-based health information dissemination [24]. The most discussed topics are the ones involving neoplastic (one manuscript) and preneoplastic disorders (two manuscripts), which still represent the most intricate and crucial challenge for the clinician [25]. Leukoplakia is a pathological entity included in the oral potentially malignant disorders, which demand clinical precision and accuracy, thus claiming the most reliable information. Differently from the other disease categories, the “autoimmune/disimmune” and the “infective” categories are the only ones that involved two pediatric topics of oral medicine, i.e., oral thrush and pediatric aphtosis. Pediatric dentistry, in fact, strongly arises as a trending topic on social platforms for all users searching the web in order to find health information for their children [26,27,28]. Lastly, we included two items in the “neuropsychiatric disorders” category, aiming to synthetize data concerning papers focused on web-based information on burning mouth syndrome and trigeminal neuralgia. The most relevant scores have been analyzed through standardized parameters such as DISCERN, GQS and quality assessment score. These tools have been widely described in the literature for the feasibility to objectify the reliability of written and videos scientific content [12,29].

### 3.1. Oral Malignant and Premalignant Disorders

#### 3.1.1. Oral Leukoplakia

In 2019, Kovalsky et al. [30] published a paper that analyzed, using Youtube^TM^’s default settings, the first 100 results obtained by typing “oral leukoplakia” on the social media platform. A total of 28 videos met the inclusion criteria with the main source labeled as “independent users or company advertisement” (n = 21) and “professional organizations or government agencies (n = 5)”. The video analysis was performed using viewer interaction, global quality scale, usefulness score, and DISCERN questionnaire. The results highlighted low quality of information, low usefulness and low reliability. The ones with more reliable information had more likes, a higher viewing rate and interaction index.

#### 3.1.2. Oral Cancer

Hassona et al. [31] evaluated the first 300 videos obtained for each research by searching the keywords “mouth cancer” and “oral cancer”. Of the 600 videos collected, only 188 met the inclusion criteria. The items found were divided into educational videos (152 videos) and testimonial videos (36 videos). The statistical analysis showed no significant correlation between video usefulness, viewing rate, viewers’ interaction and video length between the two groups. Passos et al. [32] selected 57 videos out of the 100 videos that belonged to the initial sample. The authors found that most videos were uploaded by TV channels, followed by personal profiles and health professionals, and in 70.2% of the analyzed videos, the source of information was not mentioned, and in 94.7% of the videos, there was no additional source of information. This paper highlighted no correlation between upload sources and utility scores and a weak correlation between the interaction index and the utility score.

### 3.2. Neuropsychiatric Disorders

#### 3.2.1. Burning Mouth Syndrome

Fortuna et al. [33] found that the information provided about burning mouth syndrome was of very poor quality, with the majority of videos (73.6%) scoring between 0 and 2 on a scale of 0 to 7, and less than 10% being classified as good/excellent. The overall quality of the videos was classified as poor–moderate, and there was no significant difference between the quality of videos and video length, total likes, dislikes, number of views, interaction index and views per day.

#### 3.2.2. Trigeminal Neuralgia

Wassef et al. [23] analyzed the top 20 results for each of the following six search terms: trigeminal neuralgia, trigeminal neuralgia attack, trigeminal neuralgia treatment, trigeminal neuralgia surgery, tic douloureux, and Tegretol. The keywords also included the brand name of carbamazepine for increased patient recognition. From the 120 videos obtained, only the top 10 videos by views and relevance were assessed for each of the search terms. The authors assessed the information quality using the DISCERN score (DS) and the Bias Score (BS). Further statistical analysis revealed that medical professional videos had significantly higher DS and BS compared with the ones obtained from videos uploaded by nonmedical professionals and patients.

### 3.3. Autoimmune/Disimmune Diseases

#### 3.3.1. Oral Lichen Planus (OLP)

OLP is one of the most common disorders affecting oral mucosa [34].

Morais et al. [35] evaluated the sample obtained searching the keywords “lichen planus”, “oral lichen planus” and the corresponding keywords in Portuguese and Spanish. From the 481 videos, only 37 videos were included in the statistical analysis. The results showed a significant relationship between video length and the quality and reliability of the information and the absence of association of interaction and number of views with quality or reliability. No significant differences were highlighted in quality, usefulness and reliability according to the language used. Romano et al. [36] analyzed the global quality score (GQS) and the DISCERN of the English videos resulting from searching “oral lichen planus” on the platform. The videos were also divided based on the information provider in the following categories: dentist/scientist featured, independent user, health information websites, patients featured, complementary/alternative promoting, and University channel. The authors found a statistically significant correlation of DISCERN and GQS with video length (positively) and date of upload (negatively). This results seem to suggest a slight improvement over time in medical information provided on the platform.

#### 3.3.2. Mouth Sores

Di Stasio et al. [37] involved the use of Google^TM^ Trends when analyzing the maximum search rate in the United States at the time of the paper being published; thus, the authors analyzed the first 60 videos resulting from searching for “mouth sores in children”. To assess the video quality, usefulness score (US) was used, and based on the different sources of upload, video length (VL), number of views (NV), likes (NL), dislikes (NDL), and comments (NC) were compared using ANOVA univariate and Tukey test, which showed no significant difference. Pearson’s analysis further underlined no correlation between US and every other score used.

#### 3.3.3. Sjögren Syndrome (SS)

Delli et al. [38] also used Google^TM^ Trends to identify the most suitable words to perform an electronic comprehensive search. The platform recorded the maximum search rate using the words “Sjogren’s Syndrome” without the umlaut. The first 100 videos, sorted by relevance, were included in the study sample. The videos were categorized by source of information into five groups: independent users, government/news agencies, university channels/professional organizations, health information websites, and medical advertisements/profit companies. To assess the information reliability, the authors used a modified DISCERN tool and the GQS to classify every video into three different groups: useful, misleading or personal experience. The final sample included 70 videos. Statistical analysis revealed a statistically significant difference in the mean GQS of the three different groups. The authors also emphasized that independent users predominantly uploaded videos classified as personal experience, and university channels or professional organizations predominantly uploaded useful videos.

### 3.4. Infective Diseases

#### Oral Thrush

Di Stasio et al. [39] used the same analytical method described in the previous chapter about mouth sores [37]. The search terms used were “oral thrush in children”. The data used were usefulness score (US), source of upload (SOU), video length (VL), number of views (NV), likes (NL), dislikes (NDL), and comments (NC). The final sample, after excluding the videos that did not meet the inclusion criteria, had 33 videos. The Pearson correlation revealed that US is correlated with NV, NL, and VR. The linear regression model highlighted the interdependence of US with NV, NL and VR. ANOVA and Tukey tests did not find a statistically significant difference in data between groups.

**Table 1 ijerph-19-05451-t001:** Characteristics of the 10 included studies.

	Oral Pathology	Search Strategy	Included Video (s)	Video Length	Qualitative Interpretation Tool
Delli et al. [38] (2016)	Sjögren’s syndrome	“Incognito”/“Worldwide” settings; the first 100 videos ranked by relevance were analyzed	70	5:27 ± 4:04 min	Global Quality Scale; DISCERN score; Overall content judgement
Di Stasio et al. [39] (2018)	Oral thrush	Default settings without any filters; the first 60 videos were examined	33	03:50 ± 04:58 min	Usefulness score
Di Stasio et al. [37] (2018)	Mouth sores	Default settings without any filters; the first 60 videos were examined	29	03:09 ± 01:48 min	Usefulness score
Fortuna et al. [33] (2019)	Burning mouth syndrome	Chrome incognito session; the first 10 pages of videos sorted by relevance were examined	114	142.00 s (median)	Quality assessment score
Hassona et al. [31] (2016)	Oral cancer	Default settings; the first 300 videos were examined	188	5.89 ± 5.9 min	Usefulness score
Kovalski et al. [30] (2019)	Oral leukoplakia	Default settings; the first 100 videos were examined	28	6 min 39 s (mean)	Global Quality Scale; Usefulness score; DISCERN score
Morais et al. [35] (2020)	Oral lichen planus	Setting not specified; 481 videos were examined	37	09:03 min (mean)	Usefulness score; Quality assessment score
Passos et al. [32] (2020)	Oral cancer	Default settings; the first 100 videos were examined	57	06:67 min (mean)	Utility score; DISCERN score
Romano et al. [36] (2021)	Oral lichen planus	Settings were: incognito Google Chrome, English UK Youtube^TM^ (language) and United Kingdom (country) sorting by view count, all the 215 video were examined	36	06:08 min ±338.981 s	Global Quality Scale; Quality assessment score; DISCERN score
Wassef et al. [23] (2021)	Trigeminal neuralgia	The first 120 videos, filtered by views and relevance, were examined	80	-	DISCERN score

## 4. Discussion

Oral mucosa pathologies offer a wide range of disorders which represent a demanding challenge for the clinician [40,41]; at the same time, oral disorders cover a very popular topic among affected subjects [42]. Nowadays, most of patients increasingly tend to search the web to satisfy their curiosity about their ailment [39]. For this reason, Youtube^TM^ offers a great amount of content with a medical-scientific background [31]. Oral diseases represent high-trend topics for users, so that a Youtube^TM^ query returns a large number of inherent items; nevertheless, visibility and popularity never match with valuable information [33]. The qualitative assessment of the video content is not widely homogeneous in the use of the interpretative tools: the majority of studies applied more than one tool, the most utilized ones being DISCERN [30,32,36,38] and usefulness score [30,31,35,37,39] (5 studies out of 10). Website scientific content evaluation still remains a demanding task; however, despite the assessment heterogeneity, the overall evaluation of the content quality of results is really poor, such that all included studies suggest implementation of scientific content or accurate reviews by experts in the field [33]. In fact, social platforms suffer from the lack of a peer-review process, which would allow easy and rapid dissemination of potentially uninformative content [30]. Thorny topics such as malignant disorders, such as oral cancer, require adequate information to help the patient find the most appropriate specialist in order to shorten the time for early diagnosis [31,32]. On the other hand, most videos uploaded by health professionals’ channels show high quality content, demonstrating the platform’s utility and reliability for the web community [36]. Most of the studies mentioned in this review also collected video demographic data concerning likes, dislikes, comments and number views; nevertheless, viewer’s interaction has been interpreted in a heterogeneous way by the various authors, using different tools to calculate the engagement of the user. Sharing videos represents a step into the future for dissemination of medical-scientific information: the health community should safely interact with this powerful means, broadcasting the most accurate medical content, focusing on prevention and early detection of potentially malignant disorders [30].

## 5. Conclusions

This systematic review aimed to summarize and highlight major web-disseminated oral health content retrieved on the Youtube^TM^ platform. Healthcare professionals should protect this knowledge and manage all the necessary instruments to improve web-based dissemination, in order to properly address patients’ issues about diagnosis and therapeutical pathways [35]. The main study limitations seem to be the disagreements in data extraction based on the different parameters used to analyze the content, and the unsuitability of a huge amount of non-English language video content. Moreover, there is a lack of standardization in research strategies (incognito mode, default settings, and empty cache); nevertheless, by using incognito mode, the research may produce different results based on the user’s feed and location, leading to a non-reproducible research strategy. Unfortunately, due to the dynamic nature of Youtube^TM^, the hardest challenge for the health community is represented by the peer-reviewing of misleading information and potentially dangerous news.

## Figures and Tables

**Figure 1 ijerph-19-05451-f001:**
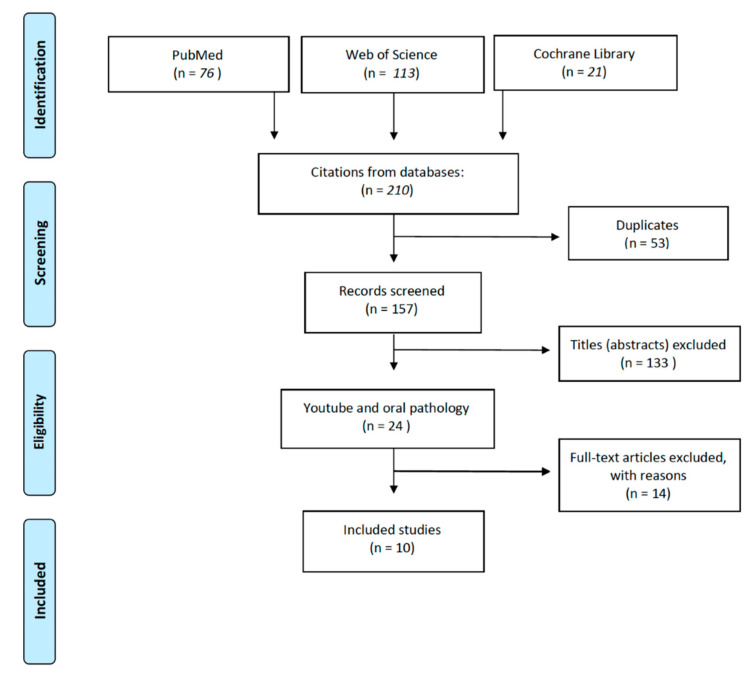
Prisma flow-chart for the selection process.

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
