# Peer review of "YoutubeTM Content Analysis as a Means of Information in Oral Medicine: A Systematic Review of the Literature"

_ijerph, 2022, doi:10.3390/ijerph19095451_

Round 1
Reviewer 1 Report
The article is interesting, raising the concern about the low quality information contained in youtube videos, and their perception by the general public. The authors followed PRISMA checklist for search, which is good.
The definition of oral medicine should be redone and citations should be redone. The authors cited themselves mostly in the article.
Grammar should be checked and corrected.

Author Response
The article is interesting, raising the concern about the low quality information contained in youtube videos, and their perception by the general public. The authors followed PRISMA checklist for search, which is good.
- The definition of oral medicine should be redone and citations should be redone. The authors cited themselves mostly in the article.
Thanks for your kind advices. As suggested by Reviewer, we have modified the definition of oral medicine, in order to clarify the issue in the Introduction section.
References section has been modified in order to remove some citations by adding new ones.
- Abukaraky, A et al. “Quality of YouTube TM videos on dental implants.” Medicina oral, patologia oral y cirugia bucalvol. 23,4 e463-e468. 1 Jul. 2018, doi:10.4317/medoral.22447
- Kodonas, Konstantinos, and Anastasia Fardi. “YouTube as a source of information about pulpotomy and pulp capping: a cross sectional reliability analysis.” Restorative dentistry & endodonticsvol. 46,3 e40. 6 Jul. 2021, doi:10.5395/rde.2021.46.e40
- Kurian, Nirmal et al. “Are YouTube videos on complete arch fixed implant-supported prostheses useful for patient education?.” The Journal of prosthetic dentistry, S0022-3913(22)00138-X. 31 Mar. 2022, doi:10.1016/j.prosdent.2022.02.013
- Charnock, D et al. “DISCERN: an instrument for judging the quality of written consumer health information on treatment choices.” Journal of epidemiology and community healthvol. 53,2 (1999): 105-11. doi:10.1136/jech.53.2.105
- Grammar should be checked and corrected.
Thanks for the suggestion; an intensive grammar check has been performed.
Reviewer 2 Report
This manuscript is a good research topic to recognize the behind and problems of YouTube, which is widely used around the world.
Also, this manuscript is considered to be a useful research topic because medicine and pharmacy-related experts need to think about the quality of scientific content.
Please make corrections by referring to the review below.
1) When was the article you searched for published?
Please write the period of the data.
The method should contain more detailed information.
2) What are the criteria to meet the abstract?
3) In the Table 1 part, it seems necessary to explain GQS, DISERN, etc. that we analyzed in previous studies.
4) Please add limitations.
Author Response
This manuscript is a good research topic to recognize the behind and problems of YouTube, which is widely used around the world.
Also, this manuscript is considered to be a useful research topic because medicine and pharmacy-related experts need to think about the quality of scientific content.
Please make corrections by referring to the review below.
- When was the article you searched for published? Please write the period of the data.
The method should contain more detailed information.
Thank you for the suggestion; authors ended search operations through the web engines on 31st of October 2021. We also clarify the Materials & Methods section, by modifying a typing error for the inclusion criteria: “the selection criteria used to include the studies were: studies on YoutubeTM videos concerning oral pathologies, available full text, and published in English since 1990.”
Then, we also deepen the Materials & Methods section in order to provide more detailed information (“Data extraction of the included papers has been synthetized in Table 1 in order to provide usable descriptive and technical information”).
- What are the criteria to meet the abstract?
Thanks for the valuable question. Following the Prisma checklist, we selected all the paper which met the inclusion criteria; we have clarified the inclusion selection of the abstracts by adding the following sentence in the Materials & Methods section: “we decided to include all the abstracts strictly related to oral pathology and medicine”.
- In the Table 1 part, it seems necessary to explain GQS, DISERN, etc. that we analyzed in previous studies.
Thank you for the suggestion. We have modified the Results section aiming to explain the GQS, DISCERN and Quality assessment score, by adding the following sentence: ”The most relevant scores have been analyzed through standardized parameters such as DISCERN, GQS and quality assessment score. These tools have been widely described in literature for the feasibility to objectify the reliability of written and videos scientific contents” reporting the related references.
- Please add limitations.
Thank you for the crucial suggestion. We have deeply modified the Conclusion section in order to highlights the main limitations that we met during the study, by adding the following paragraph: “The main study limitations seem to be the disagreements in data extraction based on the different parameters used to analyze the contents, and the unsuitability of a huge mole of non-English language video contents. Moreover there is a lack of standardization in research strategies (incognito mode, default settings, empty cache); nevertheless, also using incognito mode the research may turn different results basing on the user’s feed and location, leading to a non-reproducible research strategy”.
Reviewer 3 Report
Thank you for the opportunity to review the manuscript titled "YoutubeTM content analysis as a mean of information in oral medicine: a systematic review of the literature". As a report, this paper is well-written and interesting. The report aims to solve an important topic given the increasing amount of information available online. However, I have some recommendations which should be addressed to further improve the quality of the paper.
- Introduction. Please justify the research question. What's the research gap?
- Result. 133 out of 157 articles were excluded. This number is large. If the keywords used in this study are appropriate, we should not see such a large number. Please justify. And please also provide some examples of why some articles did not meet the inclusion criteria.
In general, this is a very practical report with great contributions. Well done!
Author Response
Thank you for the opportunity to review the manuscript titled "YoutubeTM content analysis as a mean of information in oral medicine: a systematic review of the literature". As a report, this paper is well-written and interesting. The report aims to solve an important topic given the increasing amount of information available online. However, I have some recommendations which should be addressed to further improve the quality of the paper.
- Please justify the research question. What's the research gap?
Thank you for the valuable question. We have tried to address a more specific research question modifying the introduction section by adding the following sentence “Nowadays it seems to exist a gap in knowledge regarding the reliability of health information video contents, specifically in Oral Medicine and Pathology.”
- 133 out of 157 articles were excluded. This number is large. If the keywords used in this study are appropriate, we should not see such a large number. Please justify. And please also provide some examples of why some articles did not meet the inclusion criteria.
Thank you for the valuable question. Despite the majority of items retrieved with our search strategy was related to YouTubeTM and Oral fields, the most of papers was not strictly adherent to Oral Medicine and Oral Pathology issues, but mainly on other dental subjects such as Endodontics, Prosthodontics and Implantology, that we have now referenced in the manuscript. Thanking again for the crucial advice, we have modified the Results section in order to clarify this concern by adding the following sentence: “Most of the excluded abstracts was not strictly adherent to Oral Pathology and Medicine issues since they were focused on other Stomatological fields, such as Endodontics, Prosthodontics and Implantology”.
- (Abukaraky, A et al. “Quality of YouTube TM videos on dental implants.” Medicina oral, patologia oral y cirugia bucal vol. 23,4 e463-e468. 1 Jul. 2018, doi:10.4317/medoral.22447)
- (Kodonas, Konstantinos, and Anastasia Fardi. “YouTube as a source of information about pulpotomy and pulp capping: a cross sectional reliability analysis.” Restorative dentistry & endodontics vol. 46,3 e40. 6 Jul. 2021, doi:10.5395/rde.2021.46.e40)
- (Kurian, Nirmal et al. “Are YouTube videos on complete arch fixed implant-supported prostheses useful for patient education?.” The Journal of prosthetic dentistry, S0022-3913(22)00138-X. 31 Mar. 2022, doi:10.1016/j.prosdent.2022.02.013)
In general, this is a very practical report with great contributions. Well done!
Thank you for the attention and the interest paid to the work; authors are really glad of it.